# Novel Management of Granuloma Formation Secondary to Dermal Filler with Intralesional 1444 nm Nd:YAG Laser Technique

**DOI:** 10.3390/medicina59081406

**Published:** 2023-07-31

**Authors:** Domenico Piccolo, Mohammed Hussein Mutlag, Laura Pieri, Beatrice Marina Pennati, Claudio Conforti, Paolo Bonan

**Affiliations:** 1Skin Center-Dermo Aesthetic Laser Centers, 67051 Avezzano, Italy; domenico.piccolo.skincenters@gmail.com; 2Roma Clinic, Baghdad 10001, Iraq; sales@romagroupiq.com; 3Clinical Research and Practice Department, El.En. Group, 50041 Calenzano, Italy; l.pieri@deka.it; 4Dermatology Clinic, University of Trieste, Piazza dell’Ospitale 1, 34125 Trieste, Italy; claudioconforti@yahoo.com; 5Laser Cutaneous Cosmetic & Plastic Surgery Unit, Villa Donatello Clinic, 50019 Florence, Italy; dr.pbonan@gmail.com

**Keywords:** 1444 nm Nd:YAG laser, granulomas management, dermal filler

## Abstract

*Background:* Dermal fillers for soft tissue for the treatment of face sagging, volume loss, and wrinkles have become popular among patients of all ages and ethnicities, and their use is becoming increasingly widespread. Aim: the goal of this study was to evaluate the effectiveness and safety of a micro-pulsed, 1444 nm Nd:YAG laser on dermal filler complications, in particular on granuloma management. *Methods:* A subcutaneous, 1444 nm Nd:YAG laser was used on five female patients (range age 52–68 years) with hyaluronic filler granulomas located on the face (two on the cheek area and three on the lips); three patients had self-injected the filler, buying it online. Before and after the therapy, the patients received a skin ultrasound to determine the form and location of the granulomas and to determine if there had been a full or partial resolution. During this study, all possible adverse effects at the treatment site were monitored. The 5-point Global Aesthetic Improvement Scale (GAIS) (0 point—no change; 1 point—25%, mild improvement; 2 points—50%, moderate improvement; 3 points—75%, good improvement; 4 points—100%, excellent improvement) was recorded at a 3-month follow-up. *Results:* good results were obtained in the treatment of filler granulomas with the intralesional 1444 nm laser, even if just a single treatment was performed (one intervention was effective for curing granulomas up to 5 mm in diameter). Three patients were satisfied with excellent improvement, and two patients experienced good improvement. The results are functional and aesthetically satisfying, as shown by photographic assessment. At the last follow-up, the granuloma had reduced or completely disappeared in all cases, and no infections, burns, scarring or fibrosis, episodes of severe bleeding, or other serious adverse effects had been reported. All subjects tolerated the post-treatment period well. *Conclusions:* Our findings showed that granuloma treatment with an intralesional 1444 nm Nd:YAG laser is a minimally invasive, easy, fast, efficient, and low-risk procedure.

## 1. Introduction

Dermal fillers for soft tissue for the treatment of face sagging, volume loss, and wrinkles have become popular among patients of all ages and ethnicities, and their constantly increasing widespread use has led to a greater incidence of adverse reactions, one of which is the granulomatous foreign body reaction (GFBR). There is a large variety of filler materials available: semipermanent or permanent fillers have a higher incidence of subcutaneous nodules, and therefore, they are not recommended for lip augmentation [1,2,3]. Resorbable fillers result in a low incidence of long-lasting or late complications.

Injections of permanent fillers may also result in microembolization into tiny arteries, which may end up in necrosis and possibly even acute kidney injury [4]. External compression of the nearby vasculature can potentially result in vascular impairment. The most temporary and biodegradable fillers are those based on hyaluronic acid (HA), which has a wide range of characteristics, including HA content, cross-linkage, stiffness, and elasticity [5]. The joints, eyes, and skin have the highest amounts of HA, which the body produces naturally. Because HA draws water to the skin, the skin can become more moisturized. As we age, our skin loses volume, giving us a sunken or drooping appearance on the face. To produce structure and volume on the face and lips as well as to minimize the appearance of wrinkles, fine lines, and facial creases, injectable HA is employed. Hyaluronic acid that is injected has instant effects. The use of fillers induces tissue swelling that is frequently accompanied by edema from persistent inflammatory reactions and excessive fibrosis, which can result in a GFBR and sometimes is accompanied by a bacterial infection [6]. This filler complication is significant as it has both medical and aesthetic implications, such as disfigurement or pain and dysfunction, respectively [7]. Usually, six to twelve months following injection, granuloma occurs as a subacute consequence [8]. And it is a type of nodule of inflammatory tissue that is created by aggregates of immune system cells (macrophages and T lymphocytes) and is closely associated with necrosis and scarring events (fibrosis). Granulomas are not cancerous but rather a form of chronically delayed hypersensitivity, meaning that they form after the trigger event; they are typically a response to persistent pathogens like bacteria, fungi, or corpuscular antigens that are difficult to phagocytize and contain and prevent the spread of the harmful agent to other tissues. Granuloma formation, which results from the use of dermal filler, is an inflammatory, non-allergic reaction that can be triggered by the volume of filler that is injected, filler impurities, and the filler’s physical characteristics [9]. The only non-invasive therapeutic options for granuloma secondary to dermal filler involve intralesional injection, like scar treatment, with a mixture of antibiotics (effective only if used before a biofilm has developed), steroids, fluorouracil, and hyaluronidase [10,11]. Filler manufacturers often claim that their removal can also be performed by manual aspiration but is rarely successful, especially a few months after injection, and surgical excision can be necessary. In an infected area or in granulomatous lesions, when bleeding is elevated and some scarring is unavoidable, surgery is sure to be inaccurate. In order to manage filler issues, more current improved intralesional laser procedures have been created. The 808 nm diode laser with a pulse duration of 500 ms to 1000 ms and a laser fiber with a diameter of 200 m were used by Cassuto et al. Of the 219 individuals who received treatment, 62% experienced a total removal of nodules and lumps, and 30% experienced a partial improvement. In the continuous-wave mode, Schelke et al. [12] coupled an 810 nm and a 1470 nm diode laser. They treated nodules and granulomas intralesionally in 242 patients using fibers with diameters between 200 and 600 m (penetration depth, 1 to 8 mm), and in 92% of cases, they saw improvements. Previous research has shown that the pulsed Nd:YAG laser at 1444 nm has a lipolytic action, coagulation of small vessels and reticular dermis, and is also very useful in the treatment of lipomas [13] and for removal of polyacrylamide hydrogel fillers [14]. Because both water and fat absorb the energy twice at the 1444 nm wavelength, the tissue surrounding the target region is kept safe and preserved.

Based on literature findings, the goal of this study was to estimate the effectiveness and safety of a micro-pulsed, 1444 nm Nd:YAG laser (LipoAI, Deka, Florence, Italy) on dermal filler complications, in particular on granuloma management.

## 2. Materials and Methods

### 2.1. Study Device and Patient Population

A subcutaneous, 1444 nm Nd:YAG laser (LipoAI, Deka, Florence, Italy) was used on 5 female patients (range age 52–68 years) with hyaluronic filler granulomas located on the face (2 on cheeks area and 3 on lips). Three patients had self-injected the filler, buying it online.

### 2.2. Study Protocol

All subjects gave their informed consent. All procedures were performed ambulatorily, under aseptic conditions, and after an adequate subcutaneous local anesthetic solution had been injected. After ensuring that the patient and the entire team had adequate eye protection, the laser light was applied in a fan-like pattern evenly throughout the granulomas using a 600 μm diameter optical fiber within a stainless-steel micro-cannula (max. 2 mm) and inserted in the tissue through a 1 mm incision in the skin. The fiber optic distal portion was extended about 2–3 mm past the distal end of the cannula to enable direct laser energy contact with the synthetic and organic materials. The laser’s heat output was contained to the laser tip. Heat appeared to kill bacteria, but it also seemed to melt and liquefy the microparticles. The main concept and aim of the internal laser treatment are to create several small holes in the affected region through which the granuloma’s contents are pushed to flow, facilitating the suction. The cannula can also be equipped with a thermal sensor to avoid overheating. During laser emission is suggested to continuously move the cannula on the treated area; otherwise, the risk of burns is possible. The therapy beam and the targeting beam were connected to the optical fiber from the laser head in order to view the subcutaneous laser activity while it was in operation. This trans-illumination effect decreased the risk of cutaneous burns and perforations while providing the surgeon with precise information about where and at what level the Nd:YAG laser was operating. The depth of the laser treatment (subder-mal) increases with the intensity of the targeting light. In a single session of micropulse (max. 100 s) subdermal 1444 nm Nd therapy, the following parameters were applied to the subjects: YAG laser power the power was 6 W, and the pulse rate was 30 Hz. Using a cannula and a negative pressure of 0.5 atm (50 kPa or 350–400 mmHg), the lysate (granulomatous material) was aspirated from the area after treatment. Before and after the therapy, the patients received a skin ultrasound to determine the form and location of the granulomas and to determine if there had been a full or partial resolution. Depending on the size of the granuloma, its location on the body or face, and other factors, laser therapy might take a variety of times. The temperature and specifically the resistance during the advancement of the cannula in the fibrous tissue constitute the typical endpoint for laser activity. The quantity of material removed from the patient varies according to each situation. To preserve the integrity of the skin and prevent skin damage, cold compresses were used on the laser-treated area. Sutures or sterile strips were used to seal the incision site. After the procedure, it was wrapped for a week post-operation with a smooth, non-adherent antimicrobial covering, such as compression bandages. All potential side effects at the treatment location received attention throughout the investigation. The 5-point Global Aesthetic Improvement Scale (GAIS) (0 point—no change; 1 point—25%, mild improvement; 2 points—50%, moderate improvement; 3 points—75%, good improvement; 4 points—100%, excellent improvement) was recorded by the patients at 3 months follow-up. The removal of the filler was assessed by the patient and the physician with a visual and palpation test to verify the presence of uniform tissue without any foreign body (such as granuloma residuals).

## 3. Results

We obtained good results in the treatment of filler granulomas with the intralesional 1444 nm laser, even if just a single treatment was performed (one intervention was effective for curing granulomas up to 5 mm in diameter). Three patients were satisfied with excellent improvement, and two patients experienced good improvement (see Table 1).

The technique seems to result in a much shorter healing time [15,16,17,18]. The relative delicacy of the technique also makes it eminently suitable for use in areas previously considered difficult, such as those of the face. The results are functional and aesthetically satisfying, as shown by photographic assessment (Figure 1). Using intralesional Nd:YAG laser, the necessity of surgery can be minimized. All patients were satisfied with the cannula entrance point 1 mm scars, which had disappeared a few months following laser therapy. Depending on the location of the face, slight edema was the first postoperative adverse effect and went away within 2 to 7 days. At the last follow-up, the granuloma had reduced or completely disappeared in all cases, and no infections, burns, scarring or fibrosis; episodes of severe bleeding; or other serious adverse effects had been reported. All subjects tolerated the post-treatment period well.

## 4. Discussion

Granulomatous foreign body response caused by dermal filler (GFRB) injections is generally uncommon; however, it is growing more prevalent because of the popularity of fillers and the sale of fake filler products to patients directly online [19]. A GFRB diameter may manifest as under-the-skin nodules, abscesses, and/or edema months to years after the first filler injection. With regard to the pathophysiology of a granuloma, its development in response to dermal filler injections is believed to be influenced by the filler’s chemical composition, the volume of injection, and contaminants in the product. Rapid neutrophil infiltration and the immediate binding of several host proteins to the foreign material constitute the earliest stages of granuloma development [20]. An accumulation of macrophages known as a granuloma develops in response to persistent inflammation. Granulomas are distinct from other forms of inflammation in that they develop in response to antigens that can evade neutralizing and eosinophil-specific “first-responder” inflammatory cells. Regardless of the underlying reason, all granulomas may contain extra cells and matrices. These include fibroblasts, collagen (fibrosis), eosinophils, multinucleated giant cells, lymphocytes, and neutrophils. This occurs when the immune system attempts to isolate foreign substances that it is otherwise unable to eliminate. Until now, the main medical treatments for granulomas or infectious or cystic lesions from injectable gels have been local and systemic antibiotics, steroids, fluorouracil, or hyaluronidase. The manual procedure was usually repeated once every few months to remove the thick gel. The specialists tried to make an incision, but surgery under local anesthesia in an inflammatory region may be imprecise. During the operation, there is bleeding, and there will certainly be some scarring. Laser-assisted intralesional therapy can be sufficiently effective to prevent surgery entirely. The use of a slow, easily controlled narrow laser beam allows the physician to regulate tissue damage and reduce discomfort and pain during liquefying treatment of granulomas. The present study is the first to use the intralesional 1444 nm Nd:YAG laser in analogy to laser lipolysis with a special focus on hyaluronic filler complications. It is well known that both fat and water have much larger absorption dualities at the 1444 nm wavelength. In the instance of the granuloma, the hydrogel which builds up the filler absorbs the energy. In fact, the heat generated by the laser beam can liquefy the gel and cause the capillaries and surrounding tissues to clot. Additionally, 1444 nm has the maximum ablation effectiveness when compared to the other two laser-assisted lipolysis wavelengths, 1320 nm and 1064 nm [18]. The granuloma is mechanically destroyed by the photoacoustic effect, and any infection that may be present is eliminated by the photothermal effect. As a result, the heat increases liquefaction, which subsequently makes possible drainage through the cannula. Bruising and hematomas, which are common problems following manual or surgical gel removal, can be avoided via this coagulation.

The first session of therapy was sufficient for the patients who had laser gel removed, and they did not need any more.

## 5. Conclusions

Fillers can cause a variety of adverse effects when injected into the face. Our findings showed that granuloma treatment with an intralesional 1444 nm Nd:YAG laser is a minimally invasive, easy, fast, and efficient low-risk procedure. The absence of bleeding, bruising, or hematoma development; the absence of fibrosis or scarring; and the higher likelihood of better and more liquefied gel removal are additional benefits of the laser. For all these reasons, the use of a fiber optic 1444 nm Nd:YAG laser is a safe and effective technique for hydrogel filler removal. A limitation of the study is represented by the absence of a qualitative method, such as ultrasound imaging, to assess the full removal of the filler granuloma.

## Figures and Tables

**Figure 1 medicina-59-01406-f001:**
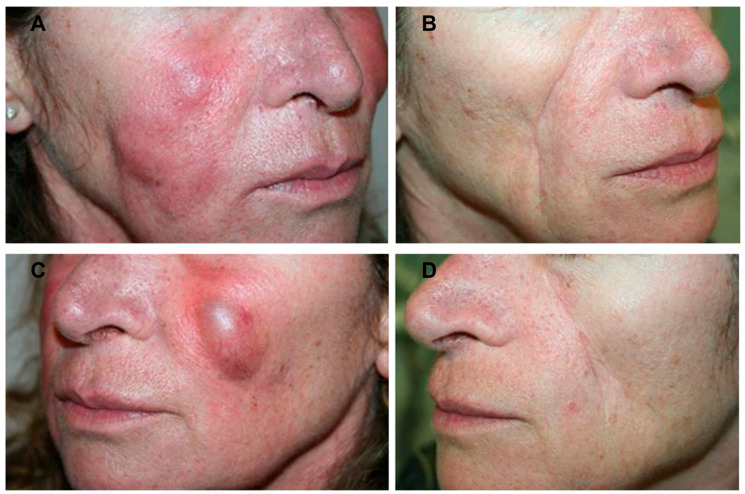
Images of a female patient who had self-injected low-cost hyaluronic acid-based fillers. Right lateral view of patient’s cheek before (**A**) and after (**B**) laser treatment. Left lateral view of the same patient’s cheek before (**C**) and after laser treatment (**D**). A complete disappearance of the granulomas was observed.

**Table 1 medicina-59-01406-t001:** The results of the 5-point Global Aesthetic Improvement Scale (GAIS) are reported for the study population.

GAIS	Number of Patients (%)
**0 points-no change**	0
**1 point-25% mild improvement**	0
**2 points-50% moderate improvement**	0
**3 points-75% good improvement**	40%
**4 points-100% excellent improvement**	60%

## Data Availability

Data are available on request due to privacy restrictions. The data presented in this study are available on request from the corresponding author.

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
