# Peer review of "Novel Management of Granuloma Formation Secondary to Dermal Filler with Intralesional 1444 nm Nd:YAG Laser Technique"

_medicina, 2023, doi:10.3390/medicina59081406_

Round 1
Reviewer 1 Report
The manuscript lacks details of the procedure and could be greatly improved with more outcome dteails and data.
Novel management of granuloma formation secondary to dermal filler with intralesional 1444 nm Nd: YAG laser technique from a
General Comments
This paper discusses removal of dermal fillers for soft tissue for the treatment of face sagging, volume loss, and wrinkles using a Nd:YAG laser. The goal of this study was to evaluate the effectiveness and safety of a micro-pulsed Nd:YAG laser on removal of granulomas that developed after HA filler treatment. During the procedure, the light is applied using an optical fiber within a stainless-steel cannula to a distance 2-3 mm past the cannula end. The progress of the procedure is monitored from an optical fiber located in the laser head. The paper implies that the granuloma was viewed using ultrasound and the authors describe a Global Aesthetic Improvement Scale to evaluate the outcomes. While this paper is interesting the authors need to provide additional valuable details about the procedure to provide the reader with enough information to understand how filler removal is achieved.
This manuscript lacks important details of the surgical procedure and outcomes, and a major revision is needed before publication is warranted.
Specific Comments
Methods
1. The manuscript needs to include a flow diagram of steps and processes used to identify the location and removal of the granulomas. How was full removal quantitatively assessed?
2. An ultrasound image of the granulomas before and after surgery would be useful to define how the exact location of the filler bulk was located and fully removed. Was filler migration noted in any of the subjects? If so, how was all the filler removed?
3. A series of images of the procedure with the location and description of the insertion and monitoring of the laser application would be useful. How was degraded filler removal assessed for completion of the process?
4. What evidence was collected that indicated that the granuloma was completely removed at the time of surgery? How long did it take to remove all filler and granulomas from each patient?
5. The authors discuss a Global Aesthetic Improvement Scale without providing clinical result data on each patient as a function of post-treatment time.
6. How long did it take for each patient to reach full cosmetic improvement? Were all the patients able to achieve 100% improvement? If not, what were the complications limiting full cosmetic improvement?
This paper would be greatly improved by adding the exact details of the procedure so the reader can follow all the necessary details to duplicate the results presented.
Author Response
- The manuscript needs to include a flow diagram of steps and processes used to identify the location and removal of the granulomas. How was full removal quantitatively assessed?
Thank you for the suggestion. An ultrasound image of the granulomas might have been used and a sentence was added in the manuscript to underline the limitation of the study. The removal of the filler was evaluated by the patients and the physicians with a visual and palpation test.
- An ultrasound image of the granulomas before and after surgery would be useful to define how the exact location of the filler bulk was located and fully removed. Was filler migration noted in any of the subjects? If so, how was all the filler removed?
Thank you for the suggestion. As mentioned before, the lack of ultrasound imaging evidence represents a limitation of the study.
- A series of images of the procedure with the location and description of the insertion and monitoring of the laser application would be useful. How was degraded filler removal assessed for completion of the process?
Thank you for the suggestion. Unfortunately, we do not have the pictures you are asking for, but we will consider adding them in future studies. The removal of the filler was evaluated by the patients and the physicians with a palpation test.
- What evidence was collected that indicated that the granuloma was completely removed at the time of surgery? How long did it take to remove all filler and granulomas from each patient?
Thank you for the question. The removal of the filler was evaluated by the patients and the physicians with a palpation test. The granuloma was removed in 1 session of 1.5 hours of treatment meaning all the procedure, from the preparation of the patient to laser treatment to the post-treatment observation.
- The authors discuss a Global Aesthetic Improvement Scale without providing clinical result data on each patient as a function of post-treatment time.
The clinical cases we report are complex and there are no standardised evaluation methods yet.
- How long did it take for each patient to reach full cosmetic improvement? Were all the patients able to achieve 100% improvement? If not, what were the complications limiting full cosmetic improvement?
Thank you for the question. Three patients involved in the study showed excellent results while two reported good results. The reason is due to the unknown components of the filler, we only know that it was a hyaluronic acid-based filler. This means that additional components are most likely involved in the inflammation process and so it was not possible to define an ad hoc intervention specific to the characteristics of every patient.
Reviewer 2 Report
The current manuscript described the successful treatment of 5 patients with dermal filler-induced granulomas by intralesional 1444 nm Nd: YAG laser. While the results appear exciting, the current format of the manuscript has to address the following issues:
Some basic information about the cases need to be included(for example, the age, location, and size of lesions). A table that lists all the essential information will be helpful. Are there any exclusion or inclusion criteria for the treatment with the 1444 nm Nd: YAG laser?
Local or transient edema or hematoma can develop immediately after the surgery and laser. How long does it take for these to resolve?
No treated patients had any significant adverse effect from the intralesional laser treatment. However, patients will some medical conditions tend to develop local bleeding, infections, or granulomas. This has to be included in the discussion.
If the 5-point global aesthetic Improvement scale (GAIS) was used for the evaluation of treatment. A detailed report of treatment results should be provided for each of the patients during the three-month follow-up. A time-course curve for each case will be preferred.
Line 141…The technique results in a much shorter healing time. The authors are expected to provide data and analysis to justify such a conclusion.
A few descriptions are not appropriate and need to be clarified. For example,
Line 27…either as a single treatment (one intervention was effective for curing granulomas up to 5 mm in diameter)
Line 110…During laser emission, is suggested to continuously move the cannula on the treated area, otherwise is a possible risk 111 of burns.
Author Response
Some basic information about the cases need to be included (for example, the age, location, and size of lesions). A table that lists all the essential information will be helpful. Are there any exclusion or inclusion criteria for the treatment with the 1444 nm Nd: YAG laser?
Thank you for the suggestion. Subjects were enrolled based on the technical characteristics reported by the device manufacturer. People reporting contraindications to the treatment were excluded from the study.
Local or transient oedema or hematoma can develop immediately after the surgery and laser. How long does it take for these to resolve?
Thank you for the question. Oedema usually resolves within 3 days from the treatment while hematoma within 5-6 days from the treatment.
No treated patients had any significant adverse effect from the intralesional laser treatment. However, patients will some medical conditions tend to develop local bleeding, infections, or granulomas. This has to be included in the discussion.
Thank you for the suggestion. A sentence was added in the discussion section.
If the 5-point global aesthetic Improvement scale (GAIS) was used for the evaluation of treatment. A detailed report of treatment results should be provided for each of the patients during the three-month follow-up. A time-course curve for each case will be preferred.
Thank you for the suggestion. We will take this into consideration for future studies for sure.
Line 141…The technique results in a much shorter healing time. The authors are expected to provide data and analysis to justify such a conclusion.
The description was improved, and references were added. We want to underline that a shorter healing time and effective performance are related to the technology we used. This because, compared to other technologies [Youn JI. A Comparison of Wavelength Dependence for Laser-assisted Lipolysis Effect Using Monte Carlo Simulation. Curr. Opt. Photon. 2009;13:267-271. https://doi.org/10.3807/JOSK.2009.13.2.267] [Tark KC, Jung JE, Song SY. Superior lipolytic effect of the 1,444 nm Nd:YAG laser: comparison with the 1,064 nm Nd:YAG laser. Lasers Surg Med. 2009 Dec;41(10):721-7. doi: 10.1002/lsm.20786. PMID: 20014250.] [Youn JI, Holcomb JD. Ablation efficiency and relative thermal confinement measurements using wavelengths 1,064, 1,320, and 1,444 nm for laser-assisted lipolysis. Lasers Med Sci. 2013 Feb;28(2):519-27. doi: 10.1007/s10103-012-1100-9. Epub 2012 Apr 26. PMID: 22534741; PMCID: PMC3586094], the combination of a microsecond impulse and a specific wavelength (1444nm) assures significant results.
Comments on the Quality of English Language
A few descriptions are not appropriate and need to be clarified. For example,
Line 27…either as a single treatment (one intervention was effective for curing granulomas up to 5 mm in diameter)
The description was improved.
Line 110…During laser emission, is suggested to continuously move the cannula on the treated area, otherwise is a possible risk 111 of burns.
The description was improved.
Reviewer 3 Report
Authors reported the ‘management of granuloma formation secondary to dermal filler using intralesional 1444 nm Nd: YAG laser’ but manuscript requires attention on following comments and same needs to be addressed in the revised manuscript:
1) Typos: Granulomatous foreign body reaction (GFBR), use standard abbreviations, discussion section used ‘GFRB’ Granulomatous foreign body response caused by dermal filler (GFRB), and it require consideration to avoid confusion to readers.
2) Result: Five patient case reports mentioned in manuscript. Representative clinical photograph of only one patient provided.
3) No detail of Global Aesthetic Improvement Scale (GAIS) of all five patients not provided, it should be provided in tabular format? Only single session of 1444 nm Nd: YAG laser given for all five patients?
4) It is very difficult to interpret the outcome of 1444 nm Nd: YAG laser in absence of GAIS score and clinical photographs for all patients.
Minor typos and reframing of sentence as per comments required.
Author Response
- Typos: Granulomatous foreign body reaction (GFBR), use standard abbreviations, discussion section used ‘GFRB’ Granulomatous foreign body response caused by dermal filler (GFRB), and it require consideration to avoid confusion to readers.
Thank you for the suggestion. The abbreviations were fixed.
- Result: Five patient case reports mentioned in manuscript. Representative clinical photograph of only one patient provided.
Thank you for the comment. It was decided to report just the most interesting and scientifically relevant case.
- No detail of Global Aesthetic Improvement Scale (GAIS) of all five patients not provided, it should be provided in tabular format? Only single session of 1444 nm Nd: YAG laser given for all five patients?
Thank you for the suggestion. A Table (now Table 1) was added in the Result section. Moreover, we confirm that a single session of 1444 nm Nd: YAG laser was given to all five patients.
- It is very difficult to interpret the outcome of 1444 nm Nd: YAG laser in absence of GAIS score and clinical photographs for all patients.
Clinical pictures were not expected in the study protocol. We decided to add a picture to the manuscript as an enrichment to the article.
Round 2
Reviewer 1 Report
The manuscript is improved and is reday for publication.
Author Response
Thank you very much for your valuable suggestions.
Reviewer 2 Report
The manuscript has been much improved after the revision. I have no more further comments to make .
The quality of English language has been improved.
Author Response

(The authors gave the same response as above.)
